# The Role of the LINC Complex in Sperm Development and Function

**DOI:** 10.3390/ijms21239058

**Published:** 2020-11-28

**Authors:** Vera Kmonickova, Michaela Frolikova, Klaus Steger, Katerina Komrskova

**Affiliations:** 1Laboratory of Reproductive Biology, Institute of Biotechnology of the Czech Academy of Sciences, BIOCEV, Prumyslova 595, 252 50 Vestec, Czech Republic; kmonickovavera@gmail.com (V.K.); Michaela.Frolikova@ibt.cas.cz (M.F.); 2Department of Urology, Pediatric Urology and Andrology, Molecular Andrology, Justus-Liebig University, 35392 Giessen, Germany; Klaus.Steger@chiru.med.uni-giessen.de; 3Department of Zoology, Faculty of Science, Charles University, Vinicna 7, 128 44 Prague 2, Czech Republic

**Keywords:** LINC complex, chromatin, nucleoskeleton, cytoskeleton, male germ cells, spermatogenesis, SUN, KASH, sperm pathologies, male fertility

## Abstract

The LINC (LInker of Nucleoskeleton and Cytoskeleton) complex is localized within the nuclear envelope and consists of SUN (Sad1/UNc84 homology domain-containing) proteins located in the inner nuclear membrane and KASH (Klarsicht/Anc1/Syne1 homology domain-containing) proteins located in the outer nuclear membrane, hence linking nuclear with cytoplasmic structures. While the nucleoplasm-facing side acts as a key player for correct pairing of homolog chromosomes and rapid chromosome movements during meiosis, the cytoplasm-facing side plays a pivotal role for sperm head development and proper acrosome formation during spermiogenesis. A further complex present in spermatozoa is involved in head-to-tail coupling. An intact LINC complex is crucial for the production of fertile sperm, as mutations in genes encoding for complex proteins are known to be associated with male subfertility in both mice and men. The present review provides a comprehensive overview on our current knowledge of LINC complex subtypes present in germ cells and its central role for male reproduction. Future studies on distinct LINC complex components are an absolute requirement to improve the diagnosis of idiopathic male factor infertility and the outcome of assisted reproduction.

## 1. Introduction

The nuclear envelope (NE) is composed of an inner nuclear membrane (INM) and an outer nuclear membrane (ONM), which are separated by a perinuclear space (PNS) that is connected with the lumen of the endoplasmic reticulum. Periodically occurring fusion sites between the INM and the ONM, known as nuclear pore complexes (NPCs), represent channels enabling exchange of molecules and information between nucleoplasm and cytoplasm [1]. While excellent reviews already exist on NE dynamics in somatic cells [2] and male germ cells [3], this review specifically focuses on the linker of nucleoskeleton and cytoskeleton (LINC) complex, as there is a growing body of evidence that LINC plays a central role for correct differentiation of male germ cells, finally resulting in the production of functional sperm. Processes requiring the participation of LINC involve chromosome movement during meiosis, as well as formation of the acrosome and shaping of the head during spermiogenesis. Within the last years, state-of-the-art techniques—such as super resolution microscopy (SRM), chromatin co-immunoprecipitation (ChIP) and the generation of knock-out mice applying CRIPR/Cas9 technology—have provided new and functional information on the interaction between various LINC components. Data suggest that a certain percentage of idiopathic male factor infertility may be caused by abnormal localization and/or aberrant composition of LINC during spermatogenesis. The present review provides a comprehensive overview on our current knowledge of LINC components and complex subtypes present in male germ cells and its central role for male reproduction. Localization and functions of involved proteins are summarized in Table 1.

## 2. LINC Complex Compositions

The LINC complex connects nuclear and cytoplasmic structures thereby enabling among other processes the transmission of cytoskeletal forces across the nuclear membrane, not only in developing cells, but also in finally differentiated cells including mature sperm [24]. Assembly of the LINC complex involves two protein families, namely (1) SUNs (Sad1/UNc84 homology domain-containing proteins) and (2) KASHs (Klarsicht/Anc1/Syne1 homology domain-containing proteins) [25]. While SUNs are located in the INM, KASHs are localized in the ONM, forming a physiological bridging structure across the NE (Figure 1).

SUNs and KASHs are type-II integral proteins comprising a single-pass transmembrane, non-cleavable, hydrophobic segment with the N-terminal end projecting into the cytoplasm [26,27,28,29]. Tethering of the LINC complex is facilitated by SUN and KASH domains (see following two chapters), which in the perinuclear space (PNS) form hydrogen bonds and interact via hydrophobic forces. In addition, formation of disulfide bonds between conserved cysteine residues has been reported [30,31]. Finally, SUN proteins comprise a coiled-coil region enabling oligomerization and providing a pocket-like structure for interaction with the highly conserved PPPT-motif present in the KASH domain [32].

The mammalian genome encodes for five SUN proteins (SUN1-5 with SUN4 also known as SPAG4) and five KASH proteins (KASH1-4, also called nesprin1-4 or SYNE1-4, and KASH5) [33]. While some proteins (SUN 1 and 2) are ubiquitously present in somatic cells, presence of others (SUN 3, 4, and 5) is restricted to the testis [14,24,26,32]. Due to various possible combinations of individual SUN and KASH proteins, different LINC complex subtypes exist each occurring within a distinct time window during germ cell development and playing a distinct role within a specific germ cell population.

For instance, in round spermatids, a LINC complex consisting of SUN1 and KASH3 appears both at the posterior pole of the nucleus and at the anterior pole on top of the acrosome. In elongating spermatids, SUN1 remains at the posterior pole close to the developing manchette, whereas a testis-specific and non-nuclear isoform of SUN1, namely SUN1ŋ lacking N-terminal sequences required for INM targeting, together with KASH3 concentrates at the anterior pole anchoring the acrosome to cytoplasmic actin filaments [8].

Another LINC complex build from SUN3 and KASH1 is located along lateral areas of the posterior nuclear pole of spermatids in close vicinity to cytoplasmic microtubules and the developing manchette [8]. As KASH1 has the potential to bind to microtubule-associated motors, such as kinesin-II and dynein–dynactin complexes, the posterior polarizing of SUN3:KASH1 LINC complex may form a bridging structure in order to connect the differentiating nucleus to surrounding manchette-associated microtubules, allowing for the transmission of cytoskeletal motor forces that drive the well-directed shaping of the spermatid head [8].

Finally, SUN4 can act as LINC complex partner for SUN3 and KASH1 [11,13] and has been suggested to be required for targeting SUN3 to the INM. As SUN4, in addition, interacts with the outer dense fiber protein-1 (ODF1), it has been hypothesized to be involved in flagellum development by linking flagellar cytoskeletal elements with central axonemal microtubules [14]. ODF1 itself represents an essential structural element of the head-to-tail coupling apparatus (HTCA) [34]. Based on its function for the HTCA [16], SUN4 may cooperate with ODF1 at the posterior nuclear pole to promote correct attachment of the spermatid head to the flagellum [15]. Another protein involved in flagellum anchoring is SUN5 [19].

### 2.1. SUN Proteins

SUN proteins are localized within the INM with its N-terminus bound to nucleoskeletal components and its C-terminus aligning to the PNS (Figure 1). The latter contains a coiled-coil region and the SUN domain mediating the link to KASH protein binding. The approximately 200 amino acids long C-terminal residue can be divided into two subdomains, namely a central α-helix and three stranded β-sheets forming a β-sandwich [29,32,35]. The α-helix occurs N-terminal to the β-sandwich and is connected by a loop-turn called KASH-lid. It has been suggested that the SUN domain forms a trimer with α-helices as the “stalk” and β-sandwiches as the “leaves” of a cloverleaf [29]. By contrast, the nucleoplasmic region of the protein is not structurally conserved and is unique to the specific function of each protein [33].

The SUN1 protein comprises 812 amino acids with proportionally equal length of the lumenal C-terminal region and the nucleoplasmic N-terminal region. Both regions are separated by a transmembrane region followed by two coiled-coil regions and the conserved SUN domain. In addition, three hydrophobic regions are present within the nucleoplasmic region [8]. In spermatocytes, SUN1 is associated with the telomere attachment sites, tethers chromosome telomeres and thus attach them to the INM playing an important role in pairing of homolog chromosomes and chromosome movement during meiosis in cooperation with KASH5 [22,23,36]. The final SUN1 transcript lacks exons 7–10, thereby splicing out the first and a part of the second hydrophobic region [8]. SUN1 is highly conserved within mammals. Mouse (*Mus musculus*) shares 90% sequence identity for the SUN1 protein with a variety of rodents, such as golden hamster (*Mesocricetus auratus*) and rat (*Rattus norvegicus*). Human (*Homo sapiens*) SUN1 reveals 90% sequence similarity with other primates, such as chimpanzee (*Pan troglodytes*) and rhesus macaque (*Macaca mulatta*). However, human SUN1 exhibits three coiled-coil regions instead of two and shares only about 60% similarity with the murine SUN1.

Interestingly, a testis-specific isoform of SUN1, known as SUN1η, exhibits a non-nuclear localization. It can be found at the apical pole of elongating and elongated spermatids being localized in the acrosomal membrane (instead of the INM) [8]. In complex with KASH3, SUN1η attaches the acrosome to cytoskeletal structures. SUN1η lacks N-terminal sequences of SUN1 that have been suggested to be involved in NE targeting [37,38].

SUN2 is a 717 amino acids long protein exhibiting two coiled-coil regions, a conserved SUN domain in the PNS and one transmembrane region. When compared with SUN1, the N-terminal region of SUN2 displays a shortening of 50 amino acid residues [24,32]. Similar to SUN1, SUN2 is present in spermatocytes and has the ability to form a complex with KASH5, thus, it may function in a similar way like SUN1. Correspondingly, SUN2 (*Mus musculus*) displays 90% sequence similarity with rat (*Rattus norvegicus*), while *Homo sapiens* shares the same percentage with chimpanzee (*Pan troglodytes*), western lowland gorilla (*Gorilla gorilla gorilla*), and bonobo (*Pan paniscus*). Human SUN2 shares 80% identity with the murine protein, however, exhibits one additional coiled-coil region.

SUN3 is an only 357 amino acids long protein with solely seven amino acids in the N-terminal nucleoplasmic domain, but a long C-terminal region in the PNS separated by a transmembrane region. The lumenal region exhibits a typical coiled-coil arrangement and contains a conserved SUN domain [8]. SUN3 forms a complex with KASH1, which then can bind to the microtubule manchette at the posterior pole (Figure 2). This complex is of pivotal importance for the formation of the sperm head during spermiogenesis [12]. SUN3 proteins of *Mus musculus*, *Homo sapiens*, and crab-eating macaque (*Macaca fascicularis)* share approximately 50% sequence identity, whereas human SUN3 shares up to 90% similarity with the western lowland gorilla (*Gorilla gorilla gorilla*) and with the rhesus macaque (*Macaca mulatta*).

SUN4 protein consists of 437 amino acids and reveals one coiled-coil region and two transmembrane regions. The N-terminal region is located in the nuclear space and the C-terminal region containing the SUN domain in the PNS [14]. SUN4 is present at three distinct localizations. Firstly, it associates with the SUN3:KASH1 complex and is involved in the head formation during spermatid differentiation (Figure 2). Secondly, it localizes to the implantation fossa, where it assists in connecting the flagellum to the head. Thirdly, it can be observed in the flagellar axoneme, where it binds to ODF-1 [14]. Mouse (*Mus musculus*) and golden hamster (*Mesocricetus auratus*) SUN4 share 90% sequence identity, whereas human SUN4 shares 100% sequence identity with chimpanzee (*Pan troglodytes*) and bonobo (*Pan paniscus*).

SUN5 protein comprises of 379 amino acids and exhibits the typical SUN protein structure including N-terminus in the nucleoplasm, C-terminus with a conserved SUN domain in the PNS, one coiled-coil domain and one transmembrane region [39]. It occurs in the INM from round spermatids to mature spermatozoa, where it gradually accumulates to the posterior pole and midpiece [19]. It is a crucial element in head-to-tail coupling. *Mus musculus*, *Homo sapiens*, pig (*Sus scrofa*), and cat (*Felis catus*) share approximately 50% sequence similarity.

### 2.2. KASH Proteins

KASH proteins, in contrast to SUN proteins, are localized in the ONM with its N-terminus oriented towards the cytoplasmic space and its C-terminus aligning to the PNS and containing the KASH domain (Figure 1). The lumenal part usually displays a short hook-like structure representing the SUN domain-binding site [33]. The KASH domain has been identified in a variety of species form yeast to human [22,23,40]. Thanks to the coiled-coil region in SUN domain proteins, it is possible for these proteins to oligomerize, specifically into trimer form. The KASH-lid of each SUN domains of the trimer binds to a single hook-like KASH domains, thus forming a 3:3 hetero-hexamer. This was proposed for LINC complexes consisting of SUN1, KASH1, and KASH2 based on the X-ray crystallographic analysis [28,31]. However, the assembly of LINC complexes containing KASH4 or KASH5 interacting with SUN1, turned out to form a complex net of 6:6 hetero-oligomeric structures that are in fact 3:3 hetero-hexamers that interact “back-to-back” with one another in the PNS [41]. Surprisingly, this 6:6 assembly was also observed with SUN1:KASH1 complex. Thus, it can be concluded that SUN-KASH complexes are rather forming a network of interacting oligomers instead of the current linear model [41]. While KASH1-4 are present in a variety of tissues, KASH5 is restricted to germ cells both in the male and in the female.

KASH1 is a large KASH domain protein with a molecular weight of approximately 1000 kDa. The former term ‘nesprin’ stands for nuclear envelope spectrin repeat because of the presence of multiple spectrin repeats in the N-terminal cytoplasmic region. The N-terminal domain is capable of binding not only the actin cytoskeleton, but also the KASH3 [33,42,43]. KASH1 forms a LINC complex with SUN3 and associates with SUN4 and is found at the posterior pole of spermatids (Figure 2). It is essential for sperm head shaping, as it can bind to the microtubule manchette via other structures. The topology of KASH1 remains the same in both *Mus musculus* and *Homo sapiens*.

KASH2 is approximately 800 kDa protein subunit of LINC complex localized in the ONM. It can be found in a variety of tissues however, its presence was not detected at any post-meiotic stages of sperm development [8]. In somatic cells, KASH2 can interact with SUN2 in the PNS and with actin filaments or microtubules in cytoplasm and plays a substantial role in nuclear polarization and migration [33].

When compared with KASH1, KASH3 is a considerably smaller protein with less spectrin repeats and with a molecular weight of 110 kDa. The characteristic KASH domain is located at the C-terminus in the PNS. In the cytoplasm, the N-terminus can bind to the intermediate filament plectin [21,33]. It creates a LINC complex with SUN1η at the apical pole of elongating and elongated spermatids and, thanks to the plectin-binding ability of KASH1, the acrosome can be connected via this LINC complex to the cytoskleleton. KASH3 topology of *Mus musculus* and *Homo sapiens* is almost the same.

KASH4 is a protein of around 42 kDa which also has not been detected at any post-meiotic stage of the sperm development [8]. Nonetheless, it was shown to be expressed in the ONM of the hair cells of the inner ear and proven to be essential for hearing [44].

Germ cell specific KASH5 protein has 648 amino acids with 20 amino acids spanning the ONM and only 20 amino acids protruding into the PNS. The majority of the protein is localized in the cytoplasm, where it exhibits a coiled-coil region. This cytoplasmic region interacts with the microtubule-associated dynein-dynactin complex [23]. KASH5 is in complex with SUN1 or SUN2 and mediates the telomere attachment to the cytoskeleton. It is essential for proper homolog pairing and chromosome movements in spermatocytes [7,22,23].

## 3. Role of the LINC Complex in Male Germ Cell Development

### 3.1. Meiotic Chromosome Movement and Telomere Attachment

Within the seminiferous epithelium, a spermatogonial stem cell undergoes several mitotic divisions and finally differentiates into primary spermatocytes, which enter meiosis. In the first meiotic division, a primary spermatocyte divides into two secondary spermatocytes. In the second meiotic division, each of the two primary spermatocytes divides again resulting in four round spermatids, which subsequently differentiate into elongated spermatids and finally—within the epididymis—into mature sperm. The first meiotic division is characterized by (1) pairing of homolog chromosomes and (2) crossing-over between sister chromatids. For correct homolog pairing and chromosome movement during meiotic prophase I, the LINC complex set up by SUN1 and KASH5 is of pivotal importance [4,22].

As SUN1 cannot directly bind to the telomere region of a chromosome, attachment of chromatin to the NE is mediated by TERB1 (TElomere Repeat-binding Bouquet formation protein-1) [45] (Figure 1A). Functional evidence has been provided by *Terb1* knockout mice demonstrating the significance of this protein for homolog pairing, chromosome movement, and telomere attachment to SUN1 and the NE [45]. TERB1 is a meiosis-specific protein, which occurs in both spermatocytes and oocytes and exhibits several domains: the N-terminal SUN1-binding domain, the C-terminal TRF1 binding (TRFB) domain and a Myb domain that bind to the meiotic-specific cohesin subunit SA3. Primarily, TERB1 associates with TERB2 (TElomere Repeat-binding Bouquet formation protein-2) and MAJIN (Membrane-Anchored JunctIoN protein) forming a multi-subunit meiotic telomere complex anchored to the INM by MAJIN, which in addition exhibits DNA-binding activity [6] (Figure 2). However, as TERB1 cannot directly bind to the telomere repeat DNA, it forms a heterocomplex with the subunit of a telomere-protecting shelterin heterocomplex (TRF1) via its TRFB domain. Formation of this heterocomplex depends on cyclin-dependent kinase (CDK) activity, because the TRFB domain contains a conserved CDK target site and, therefore, the heterocomplex formation is negatively regulated by CDK-dependent phosphorylation. Phosphorylation of the CDK target site results in a dissociation of TRF1 and finally the whole shelterin complex from the telomere. Subsequently, direct interaction between INM, LINC complex and telomere is mediated by the TERB1/2:MAJIN heterocomplex without requirement of TRF1—a process called telomere cap exchange. CDK-dependent phosphorylation also ensures a shift from prophase I to metaphase I. TERB1 has also the potential to recruit cohesin by its C-terminal region containing Myb domain and, therefore, can affect the telomere rigidity by cohesin accumulation and enrichment at meiotic telomeres [4,6,45,46].

Although SUN1 plays an important role for stable attachment of the telomere to the NE and for chromosome movement due to its linkage to cytoskeletal elements via KASH5, it has been demonstrated that the telomere:TRF1:TERB1:LINC complex connection and the telomere:TRF1:TERB1:TERB2:MAJIN:NE interaction network are two separate self-sufficient pathways that are capable of cooperation [47].

In *Sun1* knockout mice, telomere attachment is not completely lost, as some telomeres remain attached to the NE thanks to another SUN domain protein, SUN2 [7]. Of note, SUN2 cannot fully compensate for absent SUN1 in germ cells, whereas in somatic cells, SUN2 completely takes over the role of SUN1 [4,5,9]. Like SUN1, SUN2 colocalizes with KASH5 forming a LINC complex at telomere attachment sites and is involved in transmitting the cytoskeletal forces to the chromosomes. Therefore, both SUN1 and SUN2 are expressed during meiosis and colocalize at the telomere attachment site with KASH5 [7,9]. By contrast, KASH5 has the capacity to interact with the C-terminal domain of both SUN1 and SUN2 [23] suggesting the existence of a SUN1:SUN2:KASH5 heterotrimeric complex [7].

SUN1 is connected via its SUN domain to the KASH domain of KASH5 in the PNS. KASH5 is specifically expressed in both spermatocytes and oocytes and plays a fundamental role for meiotic chromosome movement due to its connection to the microtubule-associated cytoskeletal molecular motor complex, the dynein–dynactin complex [23]. Consequently, in *Kash5* knockout mice, chromosome movement is completely inhibited [36,45]. Dyneins are the largest family of molecular motor proteins and are directed to the minus-end of microtubules. Cytoplasmic dynein requires an association with a large protein complex called dynactin. KASH5 specifically binds to the p150^Glued^ subunit of dynactin with its N-terminus [48]. Thus, KASH5 acts as an adaptor for dynein. The cytoplasmic coiled-coil region of KASH5 has been proven to self-associate with another KASH5 coiled-coil region. As both SUN1 and SUN2 form a homotrimer and each trimer can bind to three KASH domains, it can be polemized that KASH5 also exhibit a trimeric structure that increases the stability of the LINC complex [22,23,49].

Rapid chromosome movements (RCMs) are a characteristic feature of meiosis. In mammals, it is a telomere-led process requiring the aforementioned potential of KASH5 to bind to the dynein–dynactin complex. RCMs culminate in a transient meiotic “bouquet” formation at the centrosome pole of the nucleus or close to the microtubule-organizing center. It is thought that it facilitates homolog pairing [50]. Using 3D fluorescence time-laps microscopy and live imaging, RCMs were demonstrated in murine spermatocytes. Interestingly, the speed of these movements differs during meiotic prophase I; it was revealed to be the lowest in S phase, followed by leptotene, RCMs accelerated but slowed down during the short bouquet stage; the speed significantly increased in zygotene and then decreased again in pachytene [36].

Besides the speed of the RCMs, trajectories seemed to be autonomous from the beginning, but appeared to be coordinated in rotation during meiotic zygotene I. During the bouquet stage, RCMs were minimal or temporarily ceased and spiraling movements of the chromosome ends were observed. Chromosomes move due to the connection with the LINC complex along stationary microtubule tracks. Lee and co-workers reported a couple of remarkable findings: (1) chromosomes can go back and forth within the same trajectory, (2) different chromosomes can move along the same trajectory in a separate timeframe, (3) the movement sometimes stops and is followed by sharp angle-turn. Microtubules associated with the NE show a distinct pattern in individual stages of meiotic prophase I and it is presumable that the arrangement of microtubule tracks somehow affects the speed and character of RCMs [36].

### 3.2. Acrosome Anchoring

In order to transmit its genetic material to the oocyte, a sperm cell must overcome (1) the *Cumulus oophorus* (an outer layer of granular cells held together by extracellular matrix) and (2) the *Zona pellucida* (a porous glycoprotein inner layer). These barriers can be passed after the so-called acrosome reaction, a regulated and specialized exocytosis of the acrosome. The acrosome reaction is species-specific, as is the shape of the acrosome resulting in a characteristic sperm head shape. Essential for the sperm head shaping is a cytoskeletal structure called the acrosome–acroplaxome–manchette complex. Acroplaxome is a thin cytoskeletal sheet containing keratin and fibrous actin surrounding the developing acrosome and determining its shape. It is also involved in anchoring the acrosome to the NE [51].

SUN1 forms a complex with KASH5 in murine meiotic germ cells, however, this is different from the situation in post-meiotic germ cells. SUN1η, a spliced variant of SUN1 lacking a part of the N-terminal domain that is crucial for INM retention [8,38], occurs in elongated spermatids and mature spermatozoa accumulates at the anterior pole of the sperm head. In contrast to SUN, SUN1η is definitely not localized within the INM, but appear to be associated with the acrosomal membrane. As SUN1η still has an intact SUN domain, it can recruit a KASH domain protein to form a LINC complex [8]. In somatic cells, KASH3 is localized within the ONM. It cannot bind directly to actin, but can bind to the cytoplasmic protein plectin, a cytoskeletal crosslinking protein, therefore establishing a connection between NE and intermediate or actin filaments [52,53]. In spermatozoa, actin is enriched in the aforementioned acroplaxome encircling the acrosome. It has been proposed that KASH3 anchors the SUN1η:KASH3 non-nuclear LINC complex via its interaction with plectin to the acroplaxome. In that way, the acrosome–acroplaxome–manchette complex is linked to the anterior pole of a cell positioning the nucleus to the right place [8].

Considering the aforementioned information, it is plausible that SUN1η represents a part of the acrosomal-membrane-system forming a pseudo-LINC complex with KASH3 and connecting the acrosome to cytoskeletal components of the acroplaxome via plectin interaction. Despite its non-nuclear localization, the SUN1η:KASH3 complex is still called LINC complex even though it does not meet the requirements for a real linker of the nucleoskeleton and cytoskeleton.

### 3.3. Nuclear Remodeling

The differentiation of round spermatids into mature spermatozoa is a highly coordinated and complex process involving a voluminous cytoplasmic structure called the microtubule manchette. It is a transient structure formed around the nuclear envelope projecting from the perinuclear ring and is constructed of parallel arrays of very stable and post-translationally modified microtubules and fibrous actin filaments. The manchette initially appears halfway through spermiogenesis at the beginning of the elongation phase and is associated with the NE until it dissembles at the end of spermiogenesis when the sperm head is fully formed. During this period, the rigid manchette moves caudally and gradually sculpts the nucleus [54,55]. The manchette does not only impact the nuclear shape, but also the formation of the flagellum as structural proteins need to be transported from the place of origin (the head) to the place of use (the tail). It is called intra-manchette and intra-flagellar transport and both are present only during spermiogenesis [56].

The manchette appears to be linked to the NE by a spermatid-specific LINC complex consisting of SUN3 in the INM and KASH1 in the ONM. The SUN3:KASH1 complex solely localizes to the posterior pole of round and elongating spermatids (Figure 1B). Interestingly, this complex covers only the lateral regions and is excluded from the very end of the posterior pole where the implantation fossa is found. This distribution is retained during the whole elongation process [8]. The connection of the manchette to the ONM is facilitated by KASH1, which like KASH5 has the ability to bind to microtubules via dynein–dynactin complex [57] or via the KIF3B subunit of kinesin II [58] or to actin with its actin-binding domain at the N-terminus. Kinesins are a superfamily of molecular motor proteins oriented towards the plus-end of microtubules and their movement is mostly powered by hydrolysis of adenosine triphosphate. However, it is still unclear which of these cytoskeletal structures is the primary one binding to the LINC complex.

Very recent work on *Sun3* knockout mice confirmed that SUN3 protein is indispensable for sperm head formation [12]. While round spermatids stay intact, consequences of SUN3 deficiency emerge during the elongation phase, where manchette microtubules fail to assemble and tether to the NE. In addition, sparse bundles of polymerized microtubules could be observed in the cytoplasm, while the perinuclear ring was not present. Absence of these structures prevents nuclear elongation and proper acrosome formation. Interestingly, SUN3 deficiency is followed by a dramatic decrease of SUN4 protein suggesting interaction of these proteins [12].

SUN4 has been identified as SUN3:KASH1 interacting partner in murine round and elongating spermatids. Like SUN3 and KASH1, SUN4 is localized to the posterior part of the NE parallel to the manchette microtubules and, at the begging, is also excluded from the implantation fossa of the nucleus (Figure 1B). In late elongating spermatids, SUN4 gradually concentrates to the very posterior end, where the sperm tail is implanted [16]. Whilst it has been demonstrated that SUN3 can easily associate with SUN4, no evidence of direct binding with KASH1 or any other KASH domain protein has been observed [11]. Pasch et al. (2015) demonstrated that SUN4 has the capacity to bind to the KASH domain of KASH1 and, in addition, can interact with itself [13]. The presumable reason for SUN3 and SUN4 interaction might be the length of the N-terminus of SUN3. Since it has a very short nucleoplasmic region, which makes interactions with the nucleoskeleton difficult, SUN4 with its larger N-terminal domain secures the integration of SUN3 to the nucleoplasmic meshwork. In the absence of SUN4, SUN3 is gravely mis-localized and strikingly could be found in the cytoplasm. KASH1 is also no longer detectable, which does not necessarily mean that localization of KASH1 is directly dependent on the presence of SUN4. Depletion of *Sun4* has also a slight impact on the distribution of the SUN1:KASH3 LINC complex [11,13]. However, to date, it remains unclear in which way SUN3 and SUN4 are interconnected.

In the human, it is almost certain that SUN4 forms a complex with SEPT12 and lamin B1 (Figure 1B). SEPT12 is a testis-specific cytoskeletal protein that belongs to an eukaryotic family of highly conserved guanosin triphosphate-binding proteins, the septins [59,60]. Lamin B1 is an intermediate filament protein scaffolding the INM. Besides providing strength and support to the NE, it also facilitates transport of molecules and helps regulating the gene expression [61]. During human spermiogenesis SEPT12, SUN4, and lamin B1 expression patterns are overlapping suggesting that they form a solid complex that is important for sperm head formation in post-meiotic germ cells. In round spermatids, SUN4, SEPT12, and lamin B1 colocalize to the nuclear periphery and gradually re-localize to the neck region with progressing elongation. In ejaculated spermatozoa, the latter proteins are fully concentrated to the midpiece region. In *SEPT12*-deficient mice, severe malformations of the sperm head could be observed. Based on expression patterns and co-immunoprecipitation findings, it can be concluded that the latter proteins establish a complex that is also pivotal for nuclear elongation and head formation [15]. Besides lamin B1, the other types of lamins can also be found in rodent and human male germ cells. Those are solely B-type lamins, specifically lamin B1 and also B3 which is the spermatid-specific isoform of lamin B2. The A-type lamins have not been detected during spermiogenesis [61]. However, there is no evidence so far that the other B-type lamins would interact with SUN proteins in germ cells.

### 3.4. Head-to-Tail Coupling

Correct linkage of the head to the tail is an absolute requirement for sperm motility. It is mediated by the head-to-tail coupling apparatus (HTCA), also called connecting piece or sperm neck. The HTCA consists of a proximal and a distal centriole, the capitulum (a dense fibrous sheet) and segmented columns (Figure 3) [62]. The capitulum is linked by bridging elements to the basal plate that is located at the implantation fossa of the NE. Two SUN domain proteins, namely SUN4 and SUN5, have been identified to be significant for the head-to-tail coupling. Even though the SUN4 is not crucial for the development of the HTCA, it is still decisive for tightening the tail anchorage to lateral parts of the basal plate [16]. In the absence of SUN5, spermatozoa are found decapitated [19].

The flagellum represents the source of sperm motility, especially the mitochondria located in the midpiece. The major part consists of the axoneme and additional structures that support elasticity and help stiffening the flagellum. These additional structures are the fibrous sheet, the mitochondrial sheet and the nine outer dense fibers (ODF), localized alongside the microtubule doublets in the axoneme. The main components of the ODFs are outer dense fiber protein-1 and 2 (ODF-1 and ODF-2, respectively). The segmented columns in the HTCA are a continuous structure of the ODF, thus containing these two main protein components, ODF-1 and ODF-2 [62]. In 1999, SUN4 has been described as an ODF-1 binding protein localized in the axoneme of the sperm flagellum [14]. This is contradictory to the current scheme that SUN domain proteins are localized to the INM. SUN4 is transcribed in round spermatids and translated in elongating spermatids exhibiting the same expression pattern like ODF-1. These two proteins interact with each other thanks to leucine zipper motifs present in both proteins, moreover, SUN4 can self-associate. Interestingly, ODF-2 also contains a leucine zipper motif, but is unable to interact with SUN4 and also no complex of ODF-2:ODF-1:SUN4 was detected. These findings prompt to assume that (1) some flanking amino acids are required for a proper interaction of SUN4 and ODF-1 and (2) elongating spermatids contains either ODF-1:ODF-2 complex or ODF-1:SUN4 complex. It has been proposed that SUN4 harbors ODF-1 molecules that can no longer bind to ODF-2 molecules, which would explain the exclusion of ODF-2 from the ODF surface facing the axoneme. The complex of SUN4:ODF-1 then localizes to the axoneme, ODF-1 forming a surface to the developing ODFs; therefore, SUN4 is engaged in ODFs positioning and acts as a molecular link between the axoneme microtubules and ODFs [14]. However, Pasch et al. (2015) never detected any immunofluorescent signal for SUN4 associated with the axoneme despite using three distinct antibodies against two different epitopes in SUN4 suggesting that SUN4 is not a part of the sperm tail [13].

In mouse, SUN4 reveals a polarized distribution pattern in the NE of testicular spermatids. Initially, it colocalizes with the transient manchette assisting in head formation and elongation, in elongating spermatids, it gradually concentrates to a posterior pole of the flagellum. In the absence of SUN4, the HTCA develops normally; however, in late elongating spermatids and epididymal spermatozoa, lateral parts of the basal plate are detached from the implantation fossa indicating that SUN4 is eminent for proper attachment of the HTCA to the NE (Figure 3) [16].

SUN5 is an essential protein in coupling sperm head to tail and—like SUN4—localizes to the HTCA. Similarly, HTCA formation remains intact, but fully detached from the implantation fossa without a functional SUN5. Detachment results in acephalic spermatozoa found in epididymal sperm [19]. A recent study indicates that SUN5 can interact with the HTCA protein DnaJ heat shock protein family (Hsp40) member B13 (DNAJB13) during spermiogenesis. DNAJB13 helps SUN5 with proper folding and promotes the connection of SUN5 with the right, yet unknown protein. Once the formation of HTCA is completed, DNAJB13 migrates from the SUN5 to the sperm tail [63]. SUN5 was originally found at the apical pole of round spermatid NE facing the acrosome suggesting a role in acrosome biogenesis and anchoring [39]. However, this theory was not corroborated by a subsequent study on *DPY19L2* knockout mice [18]. The *DPY19L2* gene encodes the transmembrane INM protein *DPY19L2* which is highly expressed in testis and basically responsible for acrosome attachment to the NE [64]. Yassine et al. (2015) demonstrated that SUN5 can be found in different cellular compartments (like Golgi apparatus membranes and the whole NE) throughout spermatogenesis before its final localization in the sperm neck in epididymal sperm. Authors also reported that when the acrosome is attached to the NE, SUN5 is excluded from this apical region, strongly indicating that SUN5 cannot be involved in the acrosome attachment to NE. In *DPY19L2* knockout mice with disrupted acrosome attachment, SUN5 spreads across the entire NE including the apical region. The fact that SUN5 attaches to the Golgi membranes indicates that it is glycosylation that functionally modifies the protein [18].

In *Sun5* knockout mice [19], abnormally shaped epididymal spermatozoa are present—the majority occur with round heads; however, also a few tailless heads can be found in the epididymis. Remarkably, round-headed spermatozoa are actually only headless motile coiled sperm tails with a cytoplasmic droplet containing disordered mitochondria, but neither acrosome, nor nucleus (Figure 4). Sperm heads are detached before entering the epididymis and remain in the epithelium of the seminiferous tubules in the so-called spermiation phase, when spermatozoa are released into the lumen of the seminiferous tubules.

When comparing wild-type with *Sun5*-deficient murine seminiferous epithelium, the only difference in the spermiation phase was that, in the wild-type, the acrosome site faced the basal membrane of the epithelium, whereas in the *Sun5*-deficient mice, the acrosome site was oriented to the lumen of the seminiferous tubules. This mis-orientation implicates that, in *Sun5*-deficient mice, heads are detached from their own tails and, therefore, cannot be properly oriented. In addition, in round spermatids, the HTCA is partially detached from the implantation fossa and, as elongation proceeds, the fragile connection between the head and tail is destroyed and only acephalic tails are released in the lumen [19]. Without SUN5, the integration of the tail to the head is completely abolished and for now, SUN5 appears to be one of the most important elements for the head-to-tail coupling.

## 4. LINC Complex and Male (In)Fertility

The mutations in genes encoding LINC complex proteins have been associated with human diseases [65] and in case of germ cells, a presence of these mutations can lead to infertility. Infertility affects one in six couples worldwide and the male partner is estimated to be involved in half of the cases. In approximately one-third of these cases, the underlying causes remain unknown [66]. Recent studies on various knock-out mice demonstrated a central role of LINC complex components for correct germ cell development and production of functional sperm (Table 1).

During meiosis, chromosomes are known to be attached to the NE by a SUN1:KASH5 complex. Consequently, both *Sun1* and *Kash5* knockout mice are infertile caused by unstable telomere attachment to the NE and faulty chromosome segregation. Male infertility in these mice is due to meiotic arrest of spermatocytes in early prophase I. As a consequence, no post-meiotic germ cells can be observed in histological tissue sections. Both knockout mouse lines develop normally into adults; however, they are sterile due to azoospermia and exhibit only small testes when compared with wild-type mice [4,22].

Small testes are also a characteristic feature of *Sun3*-deficient mice. Although these mice produce sperm, the concentration is drastically decreased and spermatozoa exhibit a globozoospermia-like phenotype with a mislocalized, fragmented, or even absent acrosome. In addition, they reveal defect flagella, since the missing SUN3 disrupts the manchette and building blocks of flagellum cannot be properly delivered resulting in insufficiently motile spermatozoa. Interestingly, *Sun3*-null mice display considerably decreased levels of SUN4 protein [12].

*Sun4*-deficient mice are also sterile due to impaired spermiogenesis. Within the seminiferous tubules, only spermatids with small nuclei with evaginated NE and uneven acrosomes are present. Due to faulty manchette assembly, there is a failure in nuclear elongating, which results in a globozoospermia-like phenotype and sterility. Lack of SUN4 is associated with an aberrant localization of the SUN1η:KASH3 complex which is followed by incorrect anchoring of the acrosome to the acroplaxome [11,13,16].

Lack of SUN5 results in the so-called acephalic spermatozoa syndrome characterized by the presence of decapitated spermatozoa. As already mentioned before, SUN5 is crucial for connecting the sperm head to tail. So far, more than 10 different mutations in the *Sun5* gene have been observed including nonsense, missense and frameshift mutations [67]. When epididymal spermatozoa of a *Sun5*-deficient mouse were evaluated for the first time, their ‘round-headed’ shape resembled those being present in globozoospermia. However, subsequent investigations identified the round ‘head’ as a cytoplasm droplet at the end of the tail containing neither chromatin, nor acrosome. Now, the condition caused by the absence of SUN5 is called pseudo-globozoospermia [19] due to the absence of a sperm head. Interestingly, the decapitated sperm cells are still motile and can move forward. Even though males with acephalic spermatozoa are categorized as infertile, the SUN5 defects can be overcome by intracytoplasmic sperm injection (ICSI), however, regular protocol when the relatively intact sperm is chosen cannot be applied in the case of *Sun5* mutant mice or patients. Due to a discovery that the sperm-like looking cells do not contain DNA, only the small fraction of tailless heads in ejaculate must be used for ICSI [19]. This strategy leads to a successful pregnancy with healthy offspring in both mouse and human [19,68].

## 5. Conclusions

Although the presence of LINC complex proteins is essential for proper functioning of somatic and germ cells and many of the mutual interactions of SUN and KASH proteins are observed in both [3], other features seem to be unique for germ cells where they were up to date observed e.g., extra-nuclear localization of a SUN isoform [65]. In recent years, it has become evident that LINC is crucial for male germ cell development into fertile sperm, as both SUN and KASH proteins represent key players for meiotic chromosome movement, nuclear positioning, acrosome formation, and sperm head shaping. Moreover, expression of some proteins—such as SUN3, SUN4, SUN5, and KASH5—has been demonstrated to be restricted to the testis (*Mus musculus*) [8,11,13,22]. While functional replacement of ubiquitously expressed proteins seems possible, e.g., SUN1 by SUN2 [7], testis-specific proteins may be irreplaceable, as lack of SUN4 [11,13] is associated with failed assembly of the microtubule manchette. As a consequence, nuclear and acrosomal formation is disrupted resulting in round-headed spermatozoa, distribution of other LINC complexes and incorrect attachment of the HTCA to the basal plate. Although an unfunctional LINC complex is not lethal, incorrect assembly of protein components is associated with male factor infertility (Table 1).

Despite reasonable endeavors that have been made during the last decade, there is still a long road ahead of us before fully understanding the functional role of the LINC complex components for male reproduction. Elucidating LINC complex interacting proteins, however, represents an important puzzle piece for understanding the overall picture of male factor infertility and for developing new strategies that, in future, may improve the outcome of assisted reproduction. To achieve this objective, quite a number of questions still have to be answered:

While the telomere attachment to the NE is already fairly well examined, some puzzle pieces are still missing, e.g., why is a significant number of chromosome telomeres still attached to the nuclear periphery in *Sun1*- and *Terb1*-deficient mice [45]? In case this is due to the presence of the MAJIN protein, which is incorporated in the INM, the question remains what happens in *Sun1*-, *Terb1*-, *Majin*-deficient mice? Are there possibly additional regulatory proteins involved in telomere attachment [6,47]? Does SUN2 only compensate for the loss of SUN1 or is it also important for correct telomere attachment? Is SUN2 able to form a complex with both KASH5 and SUN1 also in non-disrupted spermatogenesis?

Conflicting results have been reported for SUN4, which has been demonstrated to be located in the axoneme interacting with ODF-1 [14], but subsequently has been shown to be absent from the axoneme [13]. While the association of SUN4 with the HTCA [16] and its capacity to interact with ODF-1 [14] is proven, it is still a matter of debate how exactly SUN4 interacts with cytoskeletal components, since it is classified as an INM protein, hence, it is most likely that some accessory proteins are needed for spanning the ONM, probably yet unidentified KASH proteins.

Similar questions arise for the SUN5 protein, which is essential for head-to-tail coupling and also associates with the HTCA [19]. In addition, it is completely unclear how SUN4 could be localizes to the midpiece of human sperm, where no nuclear membrane is present [15].

Finally, SUN4 has been reported to be associated with the manchette, especially with the SUN3:KASH1 LINC complex. On the one hand, formation and function of the manchette is still poorly understood. On the other hand, it is so far unknown how exactly proteins of the SUN4:SUN3:KASH1 complex interact with each other. Currently, it is hypothesized that SUN3 and SUN4 may be comparable with the situation occurring for SUN1 and SUN2 forming homotrimers [31]. However, as SUN domains of SUN3 and SUN4 share only 55% sequence homology with the SUN domains of SUN1 and SUN2 [11,12], SUN3 and SUN4 may react in a different way and may form interwound heterotrimers and interact with three KASH proteins in the PNS.

## Figures and Tables

**Figure 1 ijms-21-09058-f001:**
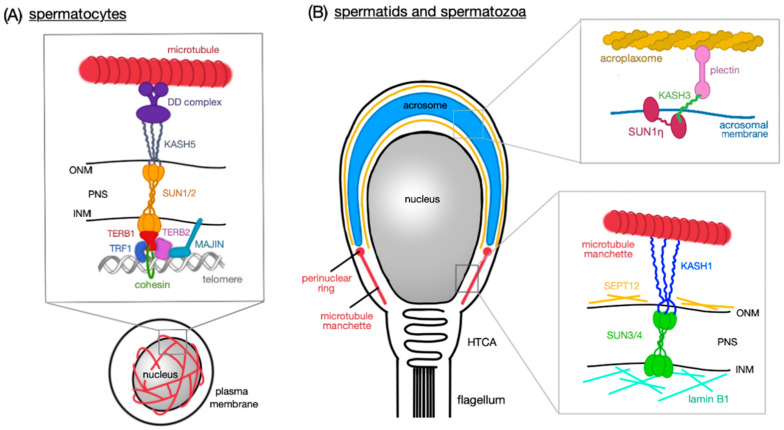
Schematic representation of LINC complex functions at different stages of sperm development. (**A**) Spermatocytes; heterotrimeric SUN1/2 forming a complex with KASH5 and enables chromosome movements during meiosis; KASH5 interacts with microtubules via dynein–dynactin (DD) complex. SUN1/2 is connected to TERB1, which recruits other nuclear proteins (TERB2, TRF1, and MAJIN) which have the capacity to bind to telomeric DNA repeats; TERB1 also binds to cohesin molecules thus stabilizing the connection between telomere and the LINC complex. (**B**) Spermatids (round and elongating—not shown) and mature spermatozoa; two distinct LINC complexes polarizes to the opposite poles of a sperm head; SUN1ŋ:KASH3 complex is associated with the acrosomal membrane at the anterior pole where it mediates the connection with the acroplaxome via plectin molecules; SUN3/4:KASH1, on the other hand, polarizes to the posterior pole of the sperm head and associates with the microtubule manchette in the cytoplasm.

**Figure 2 ijms-21-09058-f002:**
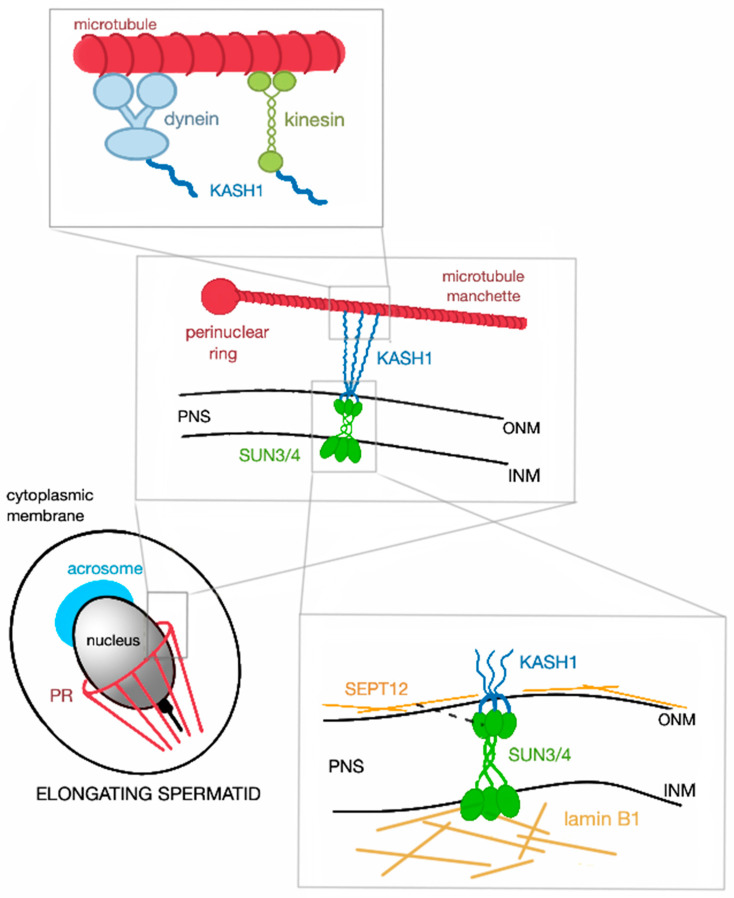
Schematic representation of SUN3/4:KASH1 LINC complex and its connection to the nucleoskeleton and cytoskeleton. In cytoplasm, KASH1 tethers the microtubule manchette to NE thanks to its ability to bind to subunits of kinesin and dynein, that can move alongside the microtubules. The nucleoplasmic part of the LINC complex, SUN3/4, is connected to lamin B1 and also associates with SEPT12 (represented by the black broken line, PR—perinuclear ring).

**Figure 3 ijms-21-09058-f003:**
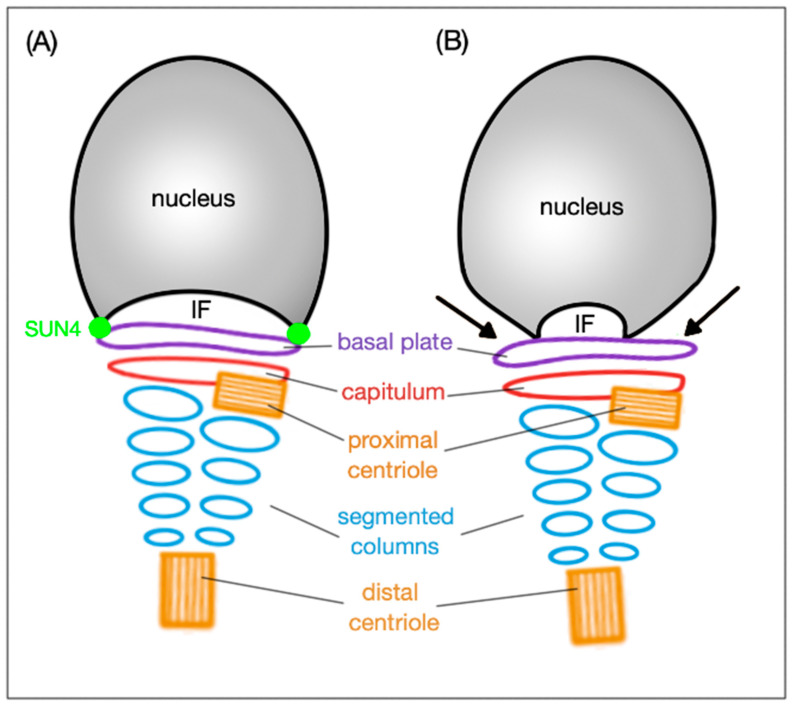
Schematic representation of the HTCA arrangement and the SUN4 significance. (**A**) In wild type, the basal plate is tightly attached to the implantation fossa (IF) thanks to the SUN4 protein. (**B**) When SUN4 is missing, the lateral regions of the basal plate are clearly detached (indicated by black arrows) from the IF; the arrangement of HTCA remains intact under both circumstances.

**Figure 4 ijms-21-09058-f004:**
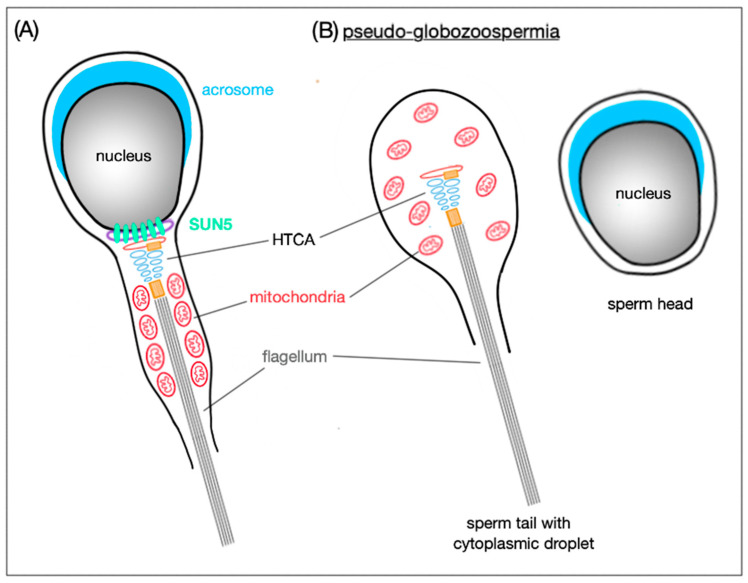
Schematic representation of SUN5 function in mature spermatozoa. (**A**) SUN5 is localized to the very posterior pole of the sperm head where it connects it to the tail. (**B**) In the absence of SUN5, the sperm head detaches completely from the sperm tail; this results in release of decapitated sperm tails with only cytoplasmic droplets in the lumen of seminiferous tubules. This phenomenon is also called pseudo-globozoospermia.

**Table 1 ijms-21-09058-t001:** Overview on localization and function of LINC complex proteins present in adult male germ cells.

LINC Protein	Germ Cell Type Subcellular Localization	Protein Function	Reproduction-Related Phenotype Due to Gene Depletion or Mutation
SUN1	spermatogonia—NE [4,5] spermatocytes—NE [4,5] round spermatids—NE [5] elongating spermatids—AL [5]	forms LINC complex with KASH5 which enables meiotic chromosome movement [4]; tethering telomeres and their attachment to the INM; in complex with TRF1, TERB1, TERB2, and MAJIN [6]; absence of SUN1 in complex SUN1:KASH5 can be compensated by SUN2 [7]	KO: sterile; small testes size; depletion of germ cells in seminiferous tubules, absence of spermatids and spermatozoa [4,5]
SUN1ŋ	round spermatids—AP [8] elongating spermatids—AP [8] elongated spermatids—AP [8]	forms LINC complex with KASH3 which binds to actin in acroplaxome via plectin; acrosome anchoring [8]	
SUN2	spermatocytes—NE [7,9]	forms LINC complex with KASH5 which enables meiotic chromosome movement; tethering telomeres and their attachment to the INM [7,9,10]	KO: fertile [10]
SUN3	round spermatids—PP [8] elongating spermatids—PP [8] elongated spermatids—PP [8]	forms LINC complex with KASH1 [8] which binds to microtubule manchette; essential for sperm head formation; associates with SUN4 [11]	KO: sterile; reduced sperm count; globozoospermia-like fenotype; defects in acrosome and flagellum [12]
SUN4	round spermatids—PP [11,13] elongating spermatids—PP [11,13] elongating spermatids—A [14] elongated spermatids—PP [11,13] elongated spermatids—A [14] mature spermatozoa—A [15]	associates with SUN3:KASH1 LINC complex; essential for sperm head formation [13]; connection of HTCA to the implantation fossa [16]; structural and organizing protein for flagellum development [12]	KO: sterile; defects in sperm head formation; globozoospermia-like phenotype [11,13,16]
SUN5	spermatocytes—NE, C [17] round spermatids—NE [18,19] elongating spermatids—PP [18,19] elongated spermatids PP, MP [18,19] mature spermatozoa—PP, MP [18,19]	inner-most anchorage of sperm tail to the nucleus [17,18,19]	KO: sterile; pseudo-globozoospermia; round spermatids: HTCA is disconnected from the implantation fossa; late spermatids: HTCA detaches completely; release of decapitated tails with cytoplasm droplet; sperm heads remain mainly in testis [18,19] HUMAN: mutation in SUN5 gene is associated with sterility and pseudo-globozoospermia [19]
KASH1	round spermatids—PP [8,13] elongating spermatids—PP [8,13] elongated spermatids—PP [8,13]	forms LINC complex with SUN3 [8]; binds to the microtubule manchette via dynein-dynactin complex or kinesin II; essential for sperm head formation; associates with SUN4 [13]	KO: fertile [20]
KASH3	elongating spermatids—AP [8] elongated spermatids—AP [8]	forms LINC complex with SUN1ŋ; binds to actin in acroplaxome via plectin; acrosome anchoring [8]	KO: fertile [21]
KASH5	spermatocytes—NE [22]	forms LINC complex with SUN1 [22,23] or SUN2 [7]; mediates the attachments of telomeres to the cytoskeletal microtubules via dynein–dynactin complex [7,22,23]	KO: sterile; small testes size (25% of WT), narrow seminiferous tubules with mainly one layer of cells, accumulation of apoptotic cells, absence of elongated spermatids and mature sperm [22]

A—axoneme; AL—acrosome-like structure; AP—apical pole; C—cytoplasm; HTCA—head-to-tail coupling apparatus; INM—inner nuclear membrane; KO—knock-out mouse model; MP—midpiece; NE—nuclear envelope; PP—posterior pole; WT—wild type.

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
