# Peer review of "The Role of the LINC Complex in Sperm Development and Function"

_ijms, 2020, doi:10.3390/ijms21239058_

Round 1
Reviewer 1 Report
The review by Kmonickova et al is a comprehensive and well written article highlighting different roles of LINC complex components with focus on germ cells.
I suggest to include some reference to diseases associated with SUN and KASH protein defects.
It is also worthy to highlight that many of the interactions of SUN and KASH proteins are observed both in somatic and germ cells, while other features have been (so far) only observed in germ cells (ex. extra-nuclear localization of a SUN isoform).
Author Response
Thank you very much for your valuable points to our manuscript. We addressed them all. Please see the response below and also the updated paper.
Lines (PDF) 484 – 485: The information and references about diseases connected with SUN and KASH protein defects in germ cells were part of manuscript. The reference focusing on diseases associated with SUN and KASH protein defects in somatic cells was added.
Méjat A, Misteli T. LINC complexes in health and disease. Nucleus. 2010;1(1):40-52. doi:10.4161/nucl.1.1.10530
Lines (PDF) 523 – 530: The information that many of the interactions of SUN and KASH proteins are observed both in somatic and germ cells, while other features have been (so far) only observed in germ cells (ex. extra-nuclear localization of a SUN isoform) was highlighted.
Reviewer 2 Report
Good review on the role of LINC complex in sperm development. Good figures.
It is a very well written paper covering all the LINC complex components comprehensively. Although, I would like the reviewers to change the title of the paper as it is misleading. There is no direct evidence cited that the LINC complex is mechanically linked to the nuclear chromatin. The evidence shows that LINC is needed for normal meiosis and chromosome movement which in turn impacts the sperm development and function. Also, the sun proteins are connected directly with the lamins in the nucleoplasm. I would recommend the authors discuss more about the connection between the SUN proteins and lamins as they have discussed about only Lamin B1.
Author Response
Thank you very much for your appreciation and valuable points to our manuscript. We addressed them. Please see the response below and also the updated paper.
- The title of paper was changed to better correspond with content of manuscript. “The role of the LINC complex in sperm development and function” is a new title of paper.
- The more information about connection between the SUN proteins and lamins were added into manuscript. See lines (PDF) 391-395.